# Spatial Organization of the Gene Regulatory Program: An Information Theoretical Approach to Breast Cancer Transcriptomics

**DOI:** 10.3390/e21020195

**Published:** 2019-02-19

**Authors:** Guillermo de Anda-Jáuregui, Jesús Espinal-Enriquez, Enrique Hernández-Lemus

**Affiliations:** 1Computational Genomics Division, National Institute of Genomic Medicine, Tlalpan 14610, Ciudad de México, Mexico; 2Centro de Ciencias de la Complejidad, Universidad Nacional Autónoma de México, Coyoacan 04510, Ciudad de México, Mexico

**Keywords:** gene regulatory program, mutual information, markov random field, spatial dependency structures, cancer transcriptomics

## Abstract

Gene regulation may be studied from an information-theoretic perspective. Gene regulatory programs are representations of the complete regulatory phenomenon associated to each biological state. In diseases such as cancer, these programs exhibit major alterations, which have been associated with the spatial organization of the genome into chromosomes. In this work, we analyze intrachromosomal, or cis-, and interchromosomal, or trans-gene regulatory programs in order to assess the differences that arise in the context of breast cancer. We find that using information theoretic approaches, it is possible to differentiate cis-and trans-regulatory programs in terms of the changes that they exhibit in the breast cancer context, indicating that in breast cancer there is a loss of trans-regulation. Finally, we use these programs to reconstruct a possible spatial relationship between chromosomes.

## 1. Introduction

### 1.1. The Gene Regulatory Program

In order to respond to external stimuli, maintain the basal function, or adapt to new environments, the cell triggers a sophisticated mechanism to produce the specific class and amount of elements responsible for carrying out the particular tasks involved in a cellular context. Processes such as development, cell differentiation and homeostasis are driven and controlled by a set of genes interacting in time and space to respond to the changing environment. We will call said set of genes and the manner in which they interact the gene regulatory program (GRP).

In a eukaryotic cell, DNA is packed in structural units called chromosomes. Human cells contain 23 chromosomes. Chromosomes are composed of DNA molecules (in which the genetic information is encoded), and structural proteins called histones, which attach the DNA molecule to them. These elements form the chromatin fiber, which in turn is coiled to generate the structure of a chromosome.

To initiate the gene transcription (production of an RNA molecule from DNA), and the consequent gene regulatory program, the chromosome must be “open”, i.e., DNA should be visible to the proteins which will carry out the transcriptional process. Opening of DNA is a highly coordinated event that allow the simultaneous production of RNA molecules in different sections of the chromosome, but also in different chromosomes. This co-regulated production of genes is one of the most important factors to generate a GRP. Fom now on, intra-chromosomal regulation will be termed cis-regulation whereas we will refer to inter-chromosomal trans-regulation. In this, we are somewhat extending or borrowing the classical concepts of cis- and trans- regulation [1] instead of the more verbose terms, intra-chromosomal and inter-chromosomal, respectively.

### 1.2. Spatial Anomalies in Cancer-Associated GRPs

The whole GRP determine the phenotype. Since gene regulation is key for the correct functioning of the living cell, abnormal performance of the way in which genes are co-regulated in time and space gives place to aberrant phenotypes. A paradigmatic example of this is cancer.

During the rise and development of a cancerous phenotype, several abnormal signals of gene regulation are triggered. This set of signals can produce faster cell growth, cell duplication and proliferation, evasion of of the immune system, and other issues. The majority of said hallmarks of cancer [2] are produced by genes in which mutations, different expression patterns or epigenetic signals appear. This altered gene expression pattern can be studied by means of next generation sequencing (NGS) techniques such as RNA-Seq, which can sequence the information from any RNA transcript from a given sample (person) at the genome-wide level.

NGS opened the possibility to have the information regarding the gene expression of the whole genome of several samples. The large data corpus allows for increases in the statistical power used, and allows the observation of the general behavior of a cancerous phenotype as compared to a non-cancerous one. Other approaches to this problem have been developed including 3C, 4C [3] and Hi-C chromosome capture techniques [4] as well as ultra-microscopy [5], among others.

With the aforementioned in mind, a simple and direct point of investigation is the observation of a GRP in cancer and the comparison with a normal (non-cancerous) program. This is, comparing at the genome-wide level, the whole set of gene interactions between these two phenotypes (cancer and normal).

Previously [6,7,8], we observed that in breast cancer, trans-(inter-chromosome) gene interactions are more scarce and weaker in cancer samples compared to the healthy phenotype. Furthermore, in breast cancer, cis-interactions become stronger, however, this is strongest between physically close genes, and this gene correlation strength decays with the distance. Said effect is not present in the normal phenotype.

In order to characterize in a quantitative manner the qualitative differences observed between the two phenotypes, in this work we have implemented an information theoretical approach, by constructing a series of indicators that, as it will be shown later, allow the classification for the distinctive patterns of both GRPs.

### 1.3. An Information Theoretical Approach to Gene Regulatory Programs

A paradigmatic question in contemporary computational biology, is the probabilistic inference of the *best* set of regulatory interactions between genes starting from a large—but incomplete—data corpus Ω. This is, being able to found the maximum-likelihood or maximum-entropy solution to the deconvolution of the GRP of the cells starting from data sampled in, say RNA sequencing experiments over whole genome transcriptomes. Such deconvolution involves the inverse problem of large scale probabilistic inference over an incomplete and noisy sample space.

A paramount solution to this extremely difficult task is founded on the tenets of information theory [9], as we will show in what follows.

Let $X_{i}=\left\{X_{1},X_{2},\ldots ,X_{N}\right\}$ be a set of *N* random variables, representing the expression levels of *N* genes in a transcriptome. For each duplex $D_{i,j}=\left(i,j\right)$ (i.e., a pair of genes), the mutual information function $I(X_{i},X_{j})$ is given by [10]:
(1)$$I\left(X_{i},X_{j}\right)=\sum_{i\in I}\sum_{j\in J}P\left(X_{i},X_{j}\right)\;\log\frac{P(X_{i},X_{j})}{P\left(X_{i}\right)\;P\left(X_{j}\right)}$$


I and J are the complete gene expression sampling spaces for genes *i* and *j* respectively—i.e., the sets of all possible values of the experimentally measured gene expression levels $X_{i}$, and $X_{j}$, within a large experimental data corpus Ω. $P(X_{i},X_{j})$ is the joint probability distribution of $X_{i}$ and $X_{j}$ and $P(X_{i})$ and $P(X_{j})$ are the marginal probability distributions of $X_{i}$ and $X_{j}$, respectively. As it is widely know, the mutual information function $I(X_{i},X_{j})$ quantifies the statistical dependence between two given random variables $X_{i}$ and $X_{j}$ [10].

We can also define the off-diagonal mutual information, $I^{†}$ as follows :
(2)$$I^{†}\left(X_{i},X_{j}\right)=I\left(X_{i},X_{j}\right)\cdot \left(1-\delta_{ij}\right)$$


$\delta_{ij}$ is Kronecker’s delta. The purpose of $I^{†}\left(X_{i},X_{j}\right)$ is to eliminate self-information from our calculations. From now on, we will drop the † superscript and we will always refer to the off-diagonal mutual information in all of our further calculations.

A GRP encompasses the full set of interactions among genes that gives rise to a transcriptional phenotype. Within the context of the theoretical and experimental settings we have just described, let us define what the solution of a GRP deconvolution problem is.

Following previous work [7], we define a gene regulatory program (GRP) as a graph $G[I\left(X_{i},X_{j}\right)]$ of all the mutual information functions for a given empirical transcriptomics sampling space Ω. It can be shown that $G[I\left(X_{i},X_{j}\right)]$ is indeed a Markov random field [11,12] considering mutual information distributions under the pairwise sufficiency assumption [13].

We will consider both cis and trans GRPs $G^{k,l}\left[I\left(X_{i},X_{j}\right)\right]$, here k,l={1,2,…,22,x} are indexes working as the chromosome label. k=l implies associations between genes *i* and *j* located in the same chromosome (cis-GRPs, $G{\left[I\left(X_{i},X_{j}\right)\right]}^{cis}$), whereas k≠l are statistical dependencies in different chromosomes (trans-GRPs, $G{\left[I\left(X_{i},X_{j}\right)\right]}^{trans}$). Partitions of the global GRP $G[I\left(X_{i},X_{j}\right)]$ into its cis- and trans- constituents, are called subregulatory programs from now on.

## 2. Analysis

### 2.1. Data

The inference of the GRPs $\;G[I\left(X_{i},X_{j}\right)]\;$ is based the RNA sequencing of basal breast cancer patients and healthy samples from The Cancer Genome Atlas (TCGA) collaboration [14] data acquired, and pre-processed as described in [6]. Briefly, we used 142 basal-like subtype breast cancer samples, as well as 101 solid-tissue normal samples. Each sample contains 15,642 annotated genes, after removal of low-counts transcripts (<5 per sample). This set of un-paired data were pre-processed, normalized and bias-reduced, to have a comparable set of expression data between cancer and normal samples.

### 2.2. GRP Inference

GRPs for tumors and controls were obtained by calculating mutual information function (MI) values for every pair of genes i,j in the genome as measured in the aforementioned RNA sequencing data. These calculations were performed using an in-house [15] parallel implementation based on the ARACNE [16] engine.

### 2.3. Measures of Change in MI between Health and Disease

In order to characterize in a quantitative manner the qualitative differences observed between the mutual information distributions making up for the statistical dependence structure behind the different conditions, we have implemented a series of indicators that, as it will be shown later allow the classification for the distinctive patterns or features of the GRPs.

Consider two GRPs $G_{tumor}^{k,l}$ and $G_{control}^{k,l}$ representing the set of interactions among genes in a phenotype. We may define a difference matrix $G_{\Delta }^{k,l}$ as follows:
$$\begin{array}{c} \;\Delta G^{k,l}=G_{tumor}^{k,l}-G_{control}^{k,l} \end{array}$$


${\Delta G}^{k,l}$ describes the changes in the interactions among genes, in terms of MI, between the two phenotypes.

#### 2.3.1. Gain Loss Score

The first indicator that we define is the gain loss score (GLS), an aggregated measure of the direction of MI changes in a GRP.
(3)$${\mathrm{GLS}}^{k,l}=\frac{|\left(\Delta \left[I\left(X_{i},X_{j}\right)\right]\in G_{\Delta }^{k,l}\right)>0\left|-\left(\Delta \left[I\left(X_{i},X_{j}\right)\right]\in G_{\Delta }^{k,l}\right)<0\right|}{\left|(\Delta \left[I\left(X_{i},X_{j}\right)\right]\in G_{\Delta }^{k,l})\right|}$$


Basically, GLS is the difference between the number of gene pairs that exhibit a gain in MI values minus the number of gene pairs that exhibit a loss in MI values between the two phenotypes, divided by the total number of gene pairs. This indicator will be positive if there are more gains, and negative if there are more losses.

#### 2.3.2. Gain Loss Ratio

The second indicator we define is the gain loss ratio (GLR), which is an aggregated measure of the magnitude of the losses and gains of MI. Basically, it is the ratio of the absolute mean value of MI gains over the absolute mean value of MI losses.
(4)$${\mathrm{GLR}}^{k,l}=\frac{absMean(\Delta G^{k,l}>0)}{absMean(\Delta G^{k,l}<0)}$$


The GLR indicator will be larger than 1 if the average value of MI gains is larger than the average value of MI losses, and will be smaller than 1 otherwise.

### 2.4. Comparison of GRPs between Control and Cancer Conditions

To assess the changes in the overall behavior of GRPs between both conditions, we used the Kolmogorov-Smirnov (KS) test. We performed the KS test between cancer $GRP_{ij}^{t}$ and control $GRP_{ij}^{c}$ GRPs to quantify the distance metric between the MI distributions. The null distribution of this statistic is calculated under the null hypothesis that the samples are drawn from the same distribution.

### 2.5. Comparison between cis- and trans-GRPs in Each Condition

To assess differences between cis-and trans-GRPs within the same biological condition, we again made use of the KS test. In each phenotype (tumor or control), we performed the KS test to compare, for each chromosome k, the difference between the cis-$GRP_{kk}$ and every trans-$GRP_{kl}$ regulatory programs.

Additionally, we decided to compare, in both biological conditions, each cis-$GRP_{kk}$ to every trans-$GRP_{kl}$ for each chromosome k by using the Hellinger distance, $H_{2}\left(X,Y\right)$. The Hellinger distance $H_{2}$ is a semi-quadratic form of f-divergence to measure the difference between two probability functions. Unlike the KS metric—already introduced– that considers maximum deviations (as given by the supremne difference), we may think of $H_{2}\left(X,Y\right)$ as a weighted average of the odds ratio given by a probability distribution X which is absolutely continuous respect to another probability distribution Y. For the case of the sub-regulatory programs we have the following expression:(5)$$H_{2}\left(\delta G^{k,k},\delta G^{k,l}\right)=\frac{1}{\sqrt{2}}\;{∥\sqrt{\delta G^{k,k}}-\sqrt{\delta G^{k,l}}∥}_{2}$$


Here ${∥\cdot ∥}_{2}$ is the Euclidean norm. $\delta G^{k,k}$ is the probability density of the cis-GRP for chromosome *k* and $\delta G^{k,l}$ is the probability density of the trans-GRP involving chromosomes *k* and *l*.

## 3. Results and Discussion

### 3.1. Intra- and Inter-Chromosome Interactions Exhibit Differences in MI Changes

We have previously observed that intra- and inter-chromosome interactions behave differently in breast cancer and regular breast tissue; if a threshold is established based on MI values, as to generate sparse graphs, the observed effect may be thought of as a loss of trans-regulation in breast cancer, as compared to healthy breast tissue [6]. By considering full cis- and trans-GRPs it is possible to further assess the way in which these types of interactions change.

In Figure 1 we observe the changes of cis-GRPs ($\Delta G^{cis}$) for every chromosome as well and trans-GRPs ($\Delta G^{trans}$) for every pair of chromosomes, in terms of two indicators: GLS, a measure of the direction of MI changes, and GLR, a measure of the magnitude of losses and gains in MI.

It may be seen that trans- interactions between any two pairs of chromosomes exhibit overall more losses than gains in terms of MI, with higher MI drops than MI gains. On the other hand, cis- interactions in each chromosome have more varied behaviors: (a) Either they also exhibit losses, but both their frequency and magnitude are lower than the one observed in trans interactions (this is the case for chromosomes 1, 2, 5, 6, 11, 17, 19 and X); or (b) they exhibit more losses than gains, but the average magnitude of the gains is higher than the average magnitude of the losses (the case for chromosomes 3, 4, 7, 8, 9, 10, 12, 13, 14, 15, 16, 18, 19, 20, and 22); the behavior of chromosome 21 is the only one where there are more gains, and gains have a higher magnitude.

GLS and GLR are proportional. As it can be observed from Figure 1, an increase in g/l scrore, is accompanied with an increase in the GLR.

### 3.2. Cis-Patterns Depend on the Chromosome Size

The structure of a chromosome is composed of two arms: The p (short) and q arms, separated by a centromere (see Appendix A). Based on the position of the centromere, the chromosomes are classified into metacentric, where the centromere is placed in the middle of the chromosome, acrocentric, where the centromere is placed closer to the extreme of the chromosome, and submetacentric, which the centromere is not in the center of the chromosome, but at neither extreme position.

There is a direct relationship between the structure and number of genes in the chromosomes: Metacentric and submetacentric chromosomes contain more genes than acrocentric chromosomes. Chromosome 1, 19, or 2, which are metacentric chromosomes, contain around 2000 genes; meanwhile chromosome 21, 22, 13 or 14 contain around 300 genes.

Interestingly, the GLS and GLR in the *cis*-GRPs exhibit a different pattern depending on the size of the chromosomes: the larger chromosomes show lower GLS and GLR than acrocentric and smaller chromosomes. Appendix A provides a more detailed description of this phenomenon.

This apparently functional behavior appears to be highly related to the structure of the chromosome. This is, during cancer the loss of information observed in terms of mutual information, depends on the number of genes in the chromosome, which is in turn related to the size of the chromosome. A possible explanation to this behavior could be related to the closeness between genes inside the chromosome. Meanwhile chromosomes 1, 17, 11 or 19 present a high density of genes, chromosomes 21, 18, or 13 are less dense and present less genes.

### 3.3. *Cis*-GRPs Are More Similar in Health and Disease than *Trans*-GRPs

Based on previous observations regarding the changes in gene regulation observed in breast cancer, the observed phenomenon may obey to one of the following: trans-regulation becoming weaker, cis-regulation becoming stronger, or a combination of the two. By comparing whole GRPs between health and disease, it is possible to have a complete assessment of this phenomenon(for complete details refer to Appendix A).

In Figure 2, a heatmap is presented in which the color intensity is proportional to the log negative Kolmogorov-Smirnov (KS) distance between GRPs in tumors and the corresponding GRPs in health $\left(-\log\left(ks_{tc}\right)\right)$. As it is kown KS distance $ks_{ij}$ arises from an uniparametric test to compare probability distributions $ks_{ij}=\sup_{x}\left|F_{i,n}\left(x\right)-F_{j,m}\left(x\right)\right|$ where $F_{i,n}$ and $F_{j,m}$ are the corresponding cummulative distributions. In the central diagonal, the KS distances between cis-GRPs may be found, while KS distances between trans-GRPs are found elsewhere. It may be seen that cis-GRPs are closer between health and disease (ranging from 0.07 to 0.18) than trans-GRPs which are notably farther.

These observations, along with those mentioned in Section 2.1, may be pointing to a phenomenon in which trans-regulation in fact becomes weaker, whereas the cis-regulation is less severely affected, and therefore prevails as the main component of the regulatory phenomenon.

### 3.4. Differences in *cis*-and *trans*-GRPs in Health and Disease

A final question is to observe whether cis-and trans-regulation behaves differently within the same phenotype. We may evaluate this through the use of GRPs. We do so by comparing, for each chromosome *k*, the $G^{k,k}$ with each $G^{k,l}$ through the use of the KS test.

In Figure 3 we show two heatmaps, one for tumors (panel A) and one for controls (panel B). In each heatmap, the color intensity is proportional to the (negative log) KS distance between the $G^{k,k}$ (cis) and the $G^{k,l}$ (trans). The figure clearly illustrates how, in the case of cancer, trans-GRPs involving a given chromosome are virtually equidistant to the corresponding cis-GRP for said chromosome. Meanwhile, in healthy breast tissue, each trans-GRP has a unique distance from the corresponding cis-GRP. Furthermore, in all chromosomes in cancer, KS values are lower than those for the healthy phenotype.

#### Reconstructing a Spatial Dimension of Gene Regulation through Information Theoretic Approaches

As we have mentioned, the difference between cis-and trans-regulation is at its core, a spatial difference, as chromosomes are fundamentally units of biological organization in localized space. Therefore, the differences observed through these information theoretical approaches may be reflecting this spatial organization.

To illustrate this, we used another distance matrix: Hellinger distance between the probability density functions (PDFs) associated to each GRP. For each chromosome *k*, the Hellinger distance between $PDF(G^{k,k})$ and each $PDF(G^{k,l})$ was calculated, in the cancer and healthy phenotypes.

In Figure 4, we show network visualizations (panel A, cancer, panel B, healthy) in which each node represents a chromosome, and the links represent the aforementioned Hellinger distance between $PDF(G^{k,k})$ and each $PDF(G^{k,l})$. Through this, we may use a force-directed layout to organize these chromosomes in space. In these visualizations, the thickness and color intensity of the edges is higher if the distance between the PDFs is smaller. Furthermore, in the case of cancer, we may observe that the layout (based on the aforementioned Hellinger distances) *pushes together* certain pairs of chromosomes (such as chromosomes 2 and X, or 3 and 8). This is a phenomenon that is not observed in healthy cells.

It is important to mention that this observation by itself is not revealing a true spatial orientation of chromosomes in the cell nucleus space. However, based on the relationship that exists between information-theoretic based correlations in gene expression, and the spatial organization of genes, it may be indicative of specific spatial arrangements observed in each phenotype. In the end, ours is a descriptive method that may serve as a hypothesis generator; ultimately, experimental validation is needed. Our results could serve as an starting point for experimental explorations using novel technologies such as hi-C, ultra-microscopy, and future related techniques.

## 4. Conclusions

Despite the enormous effort that has been devoted to dissect and analyze the molecular origins of breast cancer, the complex gene regulatory mechanisms behind this terrible disease still constitute a challenge for diagnostic and therapeutic interventions. With the advent of high throughput experimental approaches (and the big data provided by them), information theoretical tools have allowed us to analyze at an extremely detailed level such complex gene regulatory programs.

Here we have analyzed how cancer-associated gene regulatory programs present a robust phenomenon of spatial organization, associating mechanistic features of gene regulation with the three dimensional structure of genomes and its influence on the transcriptional machinery. In brief, we have discovered how the global regulatory patterns diverge from health. How some relationships are lost, and few are gained. Cis-regulation becomes the norm, while trans-regulation becomes undifferentiated. A new spatial organization thus emerges.

A number of questions and hypotheses arise from this study, namely
To what extent changes in gene regulation are relevant to breast cancer evolution?What are the possible consequences (functional or otherwise) of regulatory localization?Why different chromosomes behave differently? Including, but not limited to size effects.Are these patterns different in different cancers? Are they similar?


Rigurous quantitative studies, firmly grounded on the tenets of information theory will no doubt continue shedding light on the phenomenology of complex diseases, thus providing pivotal insights to the advancement of medical science.

## Figures and Tables

**Figure 1 entropy-21-00195-f001:**
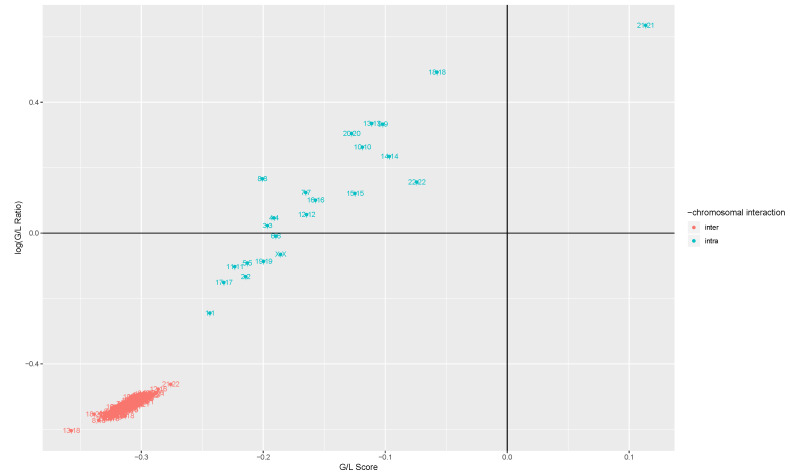
A scatterplot, where each point represents a subregulatory program for a pair of chromosomes, comprised of all mutual information function (MI) values for each pair of genes in Chromosome i and Chromosome j. By comparing the MI between gene pairs in tumor and control, in terms of gain loss score (GLS) (whether there are more losses or gains in MI) and gain loss ratio (GLR) (whether MI losses or MI gains have a higher magnitude), we identify that interchromosomal interactions between genes in any pair of chromosomes have more losses than gains of MI in disease, with an average MI loss greater than the average MI gain. Meanwhile, intrachromosomal interactions may exhibit three different behaviors: (i) they have more losses with higher average MI loss, although with higher GLS and GLR values than the interchromosomal interactions (chromosomes 1, 2, 5, 6, 11, 17, 19, X); (ii) they have more losses, but the average MI gain is higher (chromosomes 3, 4, 7, 8, 9, 10, 12, 13, 14, 15, 16, 18, 19, 20, 22) or (iii) they have more gains, with a higher average MI gain (21).

**Figure 2 entropy-21-00195-f002:**
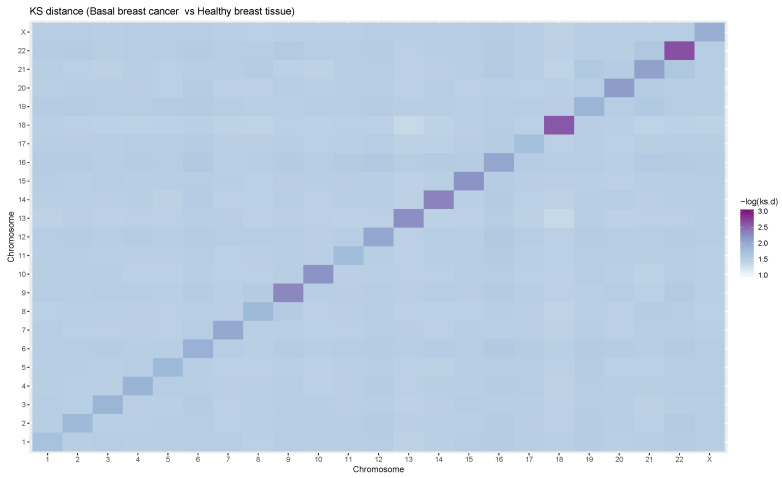
A heatmap showing the differences between GRPs in health and disease. In each square, the color intensity is proportional to $-\log(ks_{tc})$, the Kolmogorov-Smirnov (KS) distance between the subregulatory program for ij in cancer vs the subregulatory program for ij in control. We may observe that in general, the distances between trans-GRPs in control and cancer are greater than the distances between cis-GRPs in health and disease.

**Figure 3 entropy-21-00195-f003:**
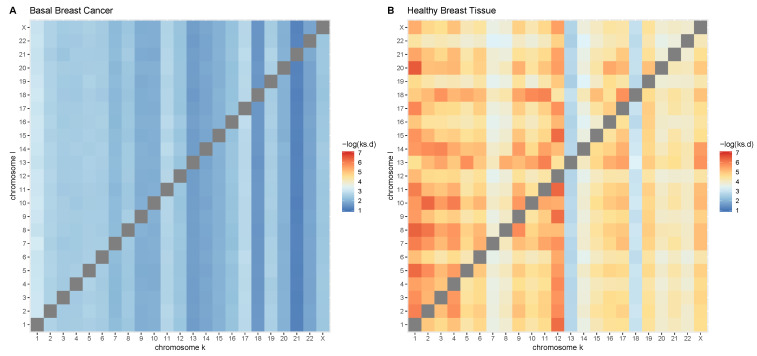
A heatmap showing the differences between cis-$GRP_{k}$ and trans-$GRP_{kl}$ in terms of KS statistic. Notice that in tumors, each trans-$GRP_{kl}$ is almost equidistant to cis-$GRP_{kk}$ (that is, each column has virtually the same color intensity in all rows), which is not the case in controls.

**Figure 4 entropy-21-00195-f004:**
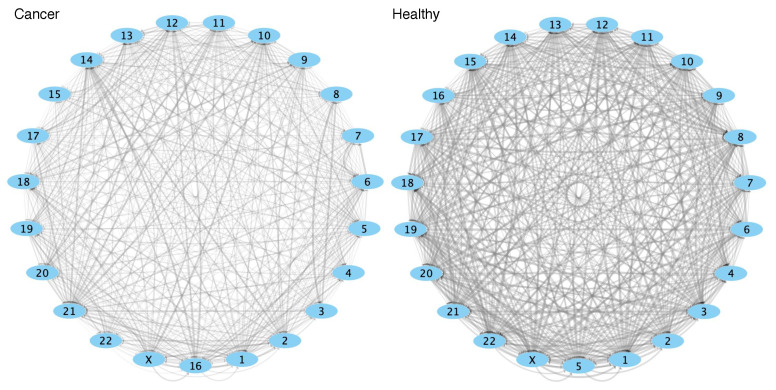
A network visualization of spatial behavior in terms of MI. Each node represents a chromosome. Each directed link has as weight the Hellinger distance (as calculated with the *textmineR* R package) between the probability density functions (PDFs) of cis-$GRP_{kk}$ and trans-$GRP_{kl}$. The intensity of each link (transparency and thickness) is inversely proportional to the Hellinger distance. The nodes are arranged using a prefuse force-directed layout algorithm, considering the inverse of the Hellinger distance. This pushes nodes where cis-$GRP_{kk}$ and trans-$GRP_{kl}$ are similar together. Notice that the position of chromosomes is different in tumors and controls. Also notice that, overall, links are thicker (that is, PDFs are closer) in controls. Appendix A provides a force-directed visualization that shows some cases in tumor networks where chromosomes are "pushed together" (such as 2 and X, or 3 and 8).

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
