# Peer review of "Spatial Organization of the Gene Regulatory Program: An Information Theoretical Approach to Breast Cancer Transcriptomics"

_entropy, 2019, doi:10.3390/e21020195_

Round 1

Reviewer 1 Report

Major comments

Title section

In my view, spatial organization seems too vague, specially considering the increasing number of studies measuring gene expression/regulation considering the spatial position of cells within issues.  Perhaps using more specific terms like spatial genome organization/chromatin organization might provide clarity.

Section 1.1, Paragraph 1

Please cite a reference supporting this claim:

“Said set of genes an the manner in which they interact is often called a Gene Regulatory Program (GRP)”

Otherwise state that you chose to refer to these genes and interactions as GRP.

Section 1.1, Paragraph 3

Please add reference supporting your explanations/definitions, or state that in this work you use cis-, trans- to discriminate intra- and inter-chromosomal events. In the regulatory genomics literature it is common to define a cis-regulatory event as one involving genomic regulatory elements acting on a target gene, as opposed to a regulatory event mediated by a trans factor such as a transcription factor. See, for example [Stergachis et. al., Nature 2014].

Section 1.2, Paragraph 3

“The large data corpus allow to increase the statistical power and observe general behavior of a cancerous phenotype and compare it with a non-cancerous one”

You mainly discuss the increase in number of variables (profiled genes -- which you refer as whole genome). These increase will actually reduce the statistical power of a cancer/control comparison -- in bulk RNAseq studies the it is usually the case that gene>>sample. If not, please clarify that the sample number is also large -- when you say “several samples” you do not convey that is the case.

Section 1.2, Paragraph 5

Although your analysis is interesting, I wonder if you deal with duplication events at all. It is known that big segmental duplications are quite common in cancer. The presence of such event will likely perturbed the measured correlations due to increase in gene counts for genes within duplicated segments in the same chromosome. As such, you should be careful with your interpretation/generalizations. In my opinion an analysis to decouple segmental duplication and patterns of correlation in your -cis, -trans view would be very interesting in itself.

Your statement that “gene-gene correlations do not depend on physical distance” is inconsistent with well-established and extensive work demonstrating otherwise [Hurst et al. 2004, NatRevGen]. In particular, due to the way transcription is coupled to chromatin accessibility (which you highlight in the introduction), RNA levels do show dependency on genomic neighborhood, which is not present at protein level -- see,  for example [Hurst et al. 2014 Hurst 2017, MolSysBio DOI 10.15252/msb.20177961; Kustatscher et al. 2017, MolSysBio DOI 10.15252/msb.20177548]. Although, this is not a direct result of the current manuscript, you should consider explaining such discrepancy, or justify your observation. Perhaps it might be related to the complexity of the tissue you are using in the cancer/non-cancer case, which is highly heterogeneous in cell-states (types), thus (in some way) averaging out  the physical effect of chromatin dynamics actually present in the cells. This, however, seems to contradict your claim of a biologically relevant intra-, inter-chromosomal differential gene correlation: i.e., the observed pattern should be a tissue level artifact, not consistent with what happens in the actual cells -- not to mention the heterogeneity of cancer tissues, both phenotypic and genomic. Please consider this observations for your interpretations/generalizations, and mention caveats or limitations of your analysis framework/datasets.

Section 2.1

Please describe the data -- e.g., number of genes, samples per group, whether case control samples come from the same individual (healthy and cancerous tissue from the same individual), etc ...

Section 2.3

Please specify the sample sizes you considered in control and tumor subjects, whether sizes are similar, or whether your approach corrects for it -- perhaps using some sort of permutation scheme.

Similarly, please show whether your measures of qualitative change has any dependency on the number of genes within each chromosome. For instance, whether in the cancer case (which might include a perturbed number of genes) you observe some bias on the observed differences relative to control in some of the pairs: high gene number vs low gene number,  high gene number vs high gene number, low gene number vs low gene number. If so, it might point to chromosomal structural changes. If your approach already accounts for this, please explain.

Section 3.2

Since the observed pattern in Fig 1 depends on chromosome size, there might be a better way to show the results -- for instance, plotting by gene size groups. I find the current figure 1 confusing. You could try to separate labels from points, indicate gene sizes or densities with point size, indicate the directionality in the Ration (i.e., high in cancer vs high in control) etc….

Related to this, please show actual gene numbers and gene densities to support your claims, and clarify gene number vs gene density observations. Chromosomes with more genes are not necessarily more gene dense, since the chromosome size difference between extremes is quite big.

Minor observations:

There is multiple typos across the text. For example: “ [cite Afraimovich]”.

I consider that the use of the “Gene regulatory program” term is unnecessary, given that what you are actually measuring is the statistical dependency, which is commonly referred to in the literature as co-expression network. Given the broad use of co-expression networks and comparisons their comparisons, I do not consider the introduction of the new term GRP justifiable.

Author Response

See reply in attached word file

Reviewer 2 Report

The authors present an application of mutual information measures in order to describe the correlation patterns between gene expression couples detecting a marked change in correlation passing by healthy to disease samples in breast cancer. Building up the observed differences in gene-pairs correlation structure they propose a possible hypothesis on 3D arrangement of chromosomes in nuclei.

manuscript is potentially very interesting and the different distribution of intra and inter chromosome gene-gene correlations are of sure interest bu there are some biological and statistical naiveties that rise some problems:

The authors propose a spatial reorganization of chromosome terriotories in the nucleus starting from disease-related changes in Mutual Information. They seem to ignore the nuclear mapping is not a tabula rasa but a well develped field thanks to the ultramicroscopy development, they could refer to any work by Cremer group (e.g. https://epigeneticsandchromatin.biomedcentral.com/articles/10.1186/s13072-017-0146-0) or any other of the vast list under the heading nuclear maps or chromatine structure the proposed disposition of the chromosomes is interestting but should be contrasted with the existing nuclear maps.

The Pearson correlation between gene expression profiles of samples coming from the same tissue kind (here breast but this holds for any tissue) being they healthy or diseased is near to unity. The authors should provide these correlations, if they are not so high probably there are some inconsistencies between heathy reference samples and diseaed one linked to the biopsies.

Point 2 makes the MI scores dominated by the need to keep the breast tissue to work as 'breast' (in other words: the breast tissue attractor is the by far dominant driving force of between genes correlation, any microarray analysis having different samples gene expression profiles as variables comes out with a first principal component explaining around 95% of total variance)...this means that reasonably the changes in MIs are noise dominated but their huge number is expected to give rise to statistical significance. 

In any case Fig.1 clarifies the 'red dots' (inter chromosome G/L) occupy the lower portion of the plot with practically no variance wirth respec to intra chromosome MI changes that are much more elevated and 'variant'. This makes us to think the most relevant part of information reside in the intrachromosome relations.

While between profile correlations are near to unity the between ggenes correlations tend to be very low, the authors should provide an estimate of the entity of between gene pairwise correlations and not only of the changes.

Figure 4 is of sure interest but I think that the main message is 'in cancer tissues there are a greater difference between cis and trans PDF MIs distribution with respect to Healthy' that is of sure interest but my dobt comes from the fact that (see Fig.1) that inter chromosome correlations are minimal and that the high number of computed correlations could provoke an artifactual change.

Author Response

See reply in attached word file

Round 2

Reviewer 2 Report

The authors now put their work in the context of state.of-the-art knowledge on nuclear maps and this is important. Moreover they clarified the 'coarse-grain' character of their approach focusing on the 'global amount of correlation' and thus (thanks to the numerosity of involved pairwise MI) skipping the actual significance of each single correlation. I am still convinced the need to maintain a largely invariant profile typical of the tissue (independent of the disaese or healthy status) and evident from the pairwise sample correlation (the transpose of the gene-gene correlation studied here) that in the present data set has a median of 0.8, imposes a strucuring of the gene-gene correlation observed by limiting the degrees of freedom, but after all this is a further issue that the authors must not necessarily consider. 

In conclusion I think that now the manuscript is worthy publication and the authors should be complimented for a very straightforward and interesting approach.